# Study on the Evolution of Deformation Trend in Laser Powder Bed Fusion Samples Based on a Novel Monitoring System

**DOI:** 10.3390/s25041043

**Published:** 2025-02-10

**Authors:** Wei Wang, Guoqing Zhang, Bing Yang, Qingpeng Chen, Zihan Yang, Sheng Liu, Fang Dong

**Affiliations:** 1School of Mechanical Science & Engineering, Huazhong University of Science and Technology, Wuhan 430074, China; m202270845@hust.edu.cn; 2The Institute of Technological Sciences, Wuhan University, Wuhan 430072, China; guoqingzhang2013@whu.edu.cn (G.Z.); qp.chen@whu.edu.cn (Q.C.); zihanyang@whu.edu.cn (Z.Y.); 3Wuhan University Shenzhen Research Institute, Shenzhen 518057, China; 4Hubei Key Laboratory of Electronic Manufacturing and Packaging Integration, Wuhan University, Wuhan 430072, China; 5School of Power and Mechanical Engineering, Wuhan University, Wuhan 430072, China; toyangbing@whu.edu.cn

**Keywords:** laser powder bed fusion, residual stress, online monitoring, mechanism of stress evolution, scanning strategy

## Abstract

During the LPBF (Laser Powder Bed Fusion) process, excessive residual stress can lead to a decline in the forming quality of printed parts. Residual stress is the primary cause of part deformation; therefore, monitoring the part deformation trend can, to a certain extent, reflect the overall residual stress level of the part. Online stress or deformation monitoring presents significant challenges due to the part being covered by powder and the need for a sealed environment. This paper proposes a novel monitoring system that replaces the bolts used to fix the baseplate with bolts capable of measuring tensile force. The deformation trend of the part will be inhibited by the bolts fixing it to the base plate, and the magnitude of the bolt tensile force is indicative of the magnitude of the deformation trend. Printing experiments were conducted under different scanning strategies and preheating conditions. The experimental curves reveal the patterns of deformation trend evolution in the parts, and these patterns are well explained by existing theories, which confirms the reliability and accuracy of the monitoring system.

## 1. Introduction

Additive manufacturing (AM) is a technology that has been widely applied in the design and production of high-performance components [1,2,3]. As AM technologies continue to evolve, metal additive manufacturing techniques, particularly Laser Powder Bed Fusion (LPBF), have been increasingly utilized in areas such as medical devices, aerospace, and the automotive industry [4]. Compared to traditional machining methods, LPBF can directly manufacture small batches of metal parts with complex structures, significantly reducing the time required for new product design and development [5,6].

In the LPBF process, substantial thermal input can lead to significant temperature gradients, resulting in mismatched elastic deformation and consequently generating high levels of residual stress within the metal structure of printed parts. Excessive residual stress can notably degrade the forming quality of the printed parts, causing issues such as deformation and cracking [7,8,9,10]. The existing studies primarily focus on optimizing printing parameters, scanning strategies, and substrate temperatures to mitigate the temperature gradients within the printed parts, thereby reducing residual stress [11,12].

Offline measurement techniques are more mature. Casavola et al. [13] utilized the hole-drilling method to measure residual stresses in parts and employed Electronic Speckle Pattern Interferometry (ESPI) to measure surface displacements caused by stress relaxation. Neutron diffraction and X-ray diffraction are also commonly used methods for measuring residual stresses [8,10,14]. Kruth et al. [15] proposed a new method, the bridge curvature method, to calculate residual stresses and investigated the effects of preheating temperature and scanning direction. Wu et al. [7] combined destructive measurement methods (hole-drilling combined with Digital Image Correlation, DIC) and non-destructive measurement methods (neutron diffraction) to evaluate residual stresses in printed parts, achieving consistent results. Through these methods, the impact of different printing parameters and substrate preheating temperatures on the residual stresses in printed parts has been assessed [11,16,17].

Various inspection techniques in additive manufacturing have been a popular topic of research [18,19,20,21,22], including studies on the in situ monitoring of residual stress and deformation in parts. Chen et al. [23] utilized Fiber Bragg Grating (FBG) sensors to monitor substrate strain during the Fused Deposition Modeling (FDM) process. Xie et al. [24] employed DIC to monitor the full-field strain of thin-walled parts during the Laser Engineered Net Shaping (LENS) process. Zeng et al. [14] used a hybrid method that integrates temperature fields, three-dimensional strain fields, and finite element simulations to obtain the stress distribution and evolution in both the substrate and the printed parts during the LENS process. Li et al. [25] proposed a Coherent Gradient Sensing (CGS) method for in situ monitoring during the Directed Energy Deposition (DED) process, which can detect the deformation evolution of the substrate at different stages.

Using numerical simulation methods to investigate the stress and deformation evolution during the printing process is also important in additive manufacturing [26]. Alzyod et al. [27] employed numerical simulation to analyze the effects of printing parameters such as printing temperature, infill density, layer thickness, and printing speed on residual stresses in parts produced by FDM. Hodge et al. [28] described a continuous modeling process for SLM, including the strong form of the problem, numerical methods, model validation, and examples. Li et al. [29,30] developed a multi-scale rapid prediction modeling method for part deformation in SLM: local residual stress fields were predicted using an equivalent heat source for meso-scale layer hatching models, and these fields were then incorporated into a macro-scale model to predict part deformation and residual stress. Denlinger et al. [31] combined a nonlinear thermo-elasto-plastic model with element coarsening strategies to simulate the thermomechanical response of large amounts of deposited material during LPBF.

In the actual production process of LPBF, the substrate and printed parts are covered by powder, making it difficult to accurately monitor the strain of the substrate using common methods such as optical images and digital images. The printing chamber is characterized by a harsh environment with high temperatures and powder flow, which imposes stringent requirements on sensors. Specifically, sensors must be capable of withstanding high temperatures, resisting powder interference, and preventing static electricity. Additionally, since the printing chamber is a sealed environment with limited space in the piston section, the sensors must be compact, easy to install, and should not compromise the chamber’s seal.

In this study, we propose a monitoring system and method for measuring substrate and part deformation trends in LPBF, called the Tension Measuring Bolt (TMB) Monitoring System. We have designed a force sensor with a bolt, which is used to monitor the tensile force produced by the substrate during the printing process, thereby reflecting the magnitude of the deformation trend in parts and the substrate. When parts are cut from the substrate, they tend to curl due to internal stresses. Kruth et al. [15] evaluated the residual stresses in parts by measuring the degree of warping. In fact, before being cut, the parts and the substrate also exhibit a certain degree of warping. This deformation trend is restrained by the bolts that fix the substrate, and measuring the tensile force of these bolts can reflect the magnitude of the deformation trend. The deformation trends during the printing process are primarily due to the thermal stresses generated in the part, which lead to residual stresses after cooling. Therefore, the magnitude of the deformation trends can reflect the overall magnitude of the residual stresses, but it does not provide information about the local residual stress.

Additionally, we have implemented a sealed structural design for the substrate to prevent contamination by metal powder particles, thereby reducing the measurement errors caused by external factors and enhancing the lifespan of the sensors. The bolt and substrate design were integrated into an existing LPBF equipment, and experiments were conducted using different scanning strategies and preheating temperatures to investigate the evolution mechanisms of deformation trend during the printing process, which were well explained from the perspective of thermal stresses and residual stresses causing deformation trend. The monitoring system proposed in this study is not only applicable to a single device but can also be used in any LPBF equipment that uses a substrate as a support for printed parts, providing new insights for the online monitoring of deformation and residual stresses in metal additive manufacturing processes.

## 2. Methods and Experimental Details

### 2.1. The TMB Monitoring System

The monitoring system consists of four TMBs, four bolt bases, and a detection end. The structural diagram of the monitoring system is shown in Figure 1a.

The structure of the bolt base is illustrated in Figure 1b. The bolt base has a circular hole with a diameter slightly larger than that of the bolt. Inside the hole, there is a metal probe connected to a spring below it, allowing the probe to be extendable and retractable. The metal probe serves a conductive function and is connected to a signal wire at the lower end of the base. Additionally, a thermocouple is designed within the base to measure the temperature of the base’s outer shell. During the printing process, the bottom of the bolt comes into contact with the metal probe. Both the bolt and the base’s outer shell are in contact with the substrate, so it is assumed that their temperatures are approximately equal.

The physical image of the TMB is shown in Figure 1c. The bottom of the TMB is coated with a metallic oxide film that exhibits piezoelectric effects.

The working principle of the TMB is illustrated in Figure 1d. The system sends a pulse signal, which, when applied to the piezoelectric film at the bottom of the bolt, is converted into a mechanical wave signal according to the inverse piezoelectric effect. This mechanical wave signal propagates through the bolt shaft. When the mechanical wave reaches the bolt head and is reflected back to the piezoelectric film, it is converted back into an electrical signal according to the piezoelectric effect. The detection end receives this signal and subtracts the time of signal transmission from the time of signal reception to obtain the time it takes for the mechanical wave to travel through the bolt shaft.

Since the length of the bolt shaft is also affected by temperature, the monitoring system is equipped with a thermocouple to measure temperature changes. This allows for the accurate calculation of the bolt shaft length at different temperatures, thereby improving the precision of the stress measurement.

In the monitoring system, two types of signals are received: Time of Flight (ToF) signals and temperature signals. The initial ToF signal $t_{0}$ and temperature signal $T_{0}$ are used as reference values. By subtracting these initial values from the measured ToF signal $t_{s}$ and temperature signal $T_{s}$, the time difference $∆t=t_{s}-t_{0}$ and the temperature difference $∆T=T_{s}-T_{0}$ can be obtained.

The elongation of the bolt consists of two components: the elongation due to tensile force and the elongation due to thermal expansion. The elongation caused by the tensile force can be calculated using Equation (1):(1)$$∆l_{F}=∆l_{sum}-∆l_{T}=v_{c}\;\cdot ∆t-l_{0}\;\cdot \alpha \;\cdot ∆T$$
where $∆l_{F}$ is the elongation due to tensile force, $∆l_{sum}$ is the total elongation, $∆l_{T}$ is the elongation due to thermal expansion, $v_{c}$ is the propagation velocity of the acoustic signal in the material, $l_{0}$ is the initial length of the material, and α is the linear thermal expansion coefficient of the material. 

Using the stress–strain relationship, the strain component $\varepsilon_{F}$ caused by the tensile force in the length direction of the bolt can be calculated as Equation (2):(2)$$\varepsilon_{F}=\frac{∆l_{F}}{l_{0}}$$

According to Hooke’s Law, the stress σ in the length direction of the bolt can be calculated as Equation (3):(3)$$\sigma =E\cdot \varepsilon_{F}=E\cdot \frac{∆l_{F}}{l_{0}}$$
where E is the Young’s modulus of the material. Given that the linear thermal expansion coefficient α is very small, the change in the cross-sectional area of the bolt can be neglected. Therefore, the tensile force F acting on the bolt can be given by the following equation:(4)$$F=\sigma \cdot S\approx \;\sigma \cdot S_{0}=\frac{ES_{0}v_{c}}{l_{0}}\cdot ∆t-ES_{0}\alpha \cdot \;∆T=B\cdot ∆t-K\cdot ∆T$$
where $S_{0}$ is the initial cross-sectional area of the bolt. Since E, $S_{0}$, $v_{c}$, $l_{0}$, and α are constants, the coefficients before each variable can be represented by constants B and K, thus simplifying the equation to F=B·∆t−K·∆T. The coefficient B is referred to as the load coefficient, and the coefficient K is referred to as the temperature coefficient. The values of these coefficients are related to the material and dimensions of the bolt. This equation can be used to calculate the magnitude of the tensile force exerted on the bolt by external factors.

For each processed bolt, calibration must be performed before use. By using a tensile testing machine to conduct tensile tests without changing the temperature, the load coefficient B can be determined. In the absence of external force, the relationship between the ToF signal difference ∆t and the temperature difference ∆T can be measured to determine the temperature coefficient K. Once the bolt is calibrated and integrated into the TMB Monitoring System, the tensile force acting on the bolt can be precisely calculated using Equation (4). For the ∅6 × 25 mm bolts used in our experiment, the precision can reach 0.01 kN. The tensile force measured using this method is independent of temperature, so the effects of temperature changes will not be discussed in the subsequent conclusions and discussions.

Figure 1a shows a schematic diagram of the structure when the monitoring system is in operation, which includes the following: A support plate fixed to the main body of the equipment. A substrate set on the support plate for sintering and forming printed parts. Four TMBs used to fix the substrate to the support plate. A monitoring terminal located near the printing equipment, which includes a data acquisition card for the TMBs and a data acquisition card for the thermocouple used to measure temperature, connected to the bolt base by wires.

During the printing process, significant thermal stresses can cause the printed part to deform, and this deformation is transferred to the substrate. The deformation of the substrate is restrained by the TMBs and manifests as axial forces on the bolts. The monitoring components detect the deformation of the TMBs through the data acquisition device and output the magnitude of the tensile force acting on the TMBs as the signal value.

Here, presents the detailed steps for fabricating TMB:**Surface Cleaning and Preparation:** First, the bottom face of TMB is cleaned using an ultrasonic cleaning device to remove any dirt, oil, or other contaminants. Cleaning is further enhanced by using high-purity solvents such as ethanol or acetone to ensure that the surface is free of any residues. Depending on the requirements, the bottom face of TMB may be subjected to mechanical or electrochemical polishing to improve its flatness and smoothness. The primary goal is to minimize surface roughness, thereby ensuring uniformity and adhesion of the subsequent coating.Fixation and Vacuum Setup: The bolt is securely fastened to the worktable inside the arc ion plating chamber, ensuring that the bottom face of TMB is exposed and can receive the coating uniformly. A vacuum pump is then activated to reduce the pressure in the chamber to the specified vacuum level, typically below 10-5 Torr. This step is crucial to eliminate air and moisture from the plating environment, preventing any contaminants from interfering with the coating process.Arc Ion Source Activation: The arc ion source is activated, and the arc is ignited and stabilized. The arc discharge generates a high-energy ion beam, which provides ZnO (zinc oxide) material ions for the coating. The arc current, voltage, and gas flow rates (such as oxygen and argon) are adjusted to optimize the arc stability and coating quality. Typically, oxygen acts as a reactive gas in the formation of ZnO, while argon serves as an inert gas to maintain the arc.Deposition of ZnO Coating: A high-purity Zn (zinc) target is placed in the target position of the arc ion source. The arc discharge causes the Zn target to vaporize and ionize, generating a Zn ion beam. The Zn ions deposit on the bottom face of TMB and react with oxygen in the environment to form a ZnO coating. The deposition time and ion flux density are controlled to ensure that the coating thickness and uniformity meet the design specifications. The arc discharge and ion deposition continue until the ZnO coating reaches the desired thickness, usually in the micrometer range. The growth process of the coating is monitored in real-time to adjust deposition parameters and ensure the quality of the coating.Cooling and Post-Processing: After the deposition process is completed, the bottom face of TMB is allowed to cool gradually in a vacuum to prevent the thermal stress caused by sudden cooling. The vacuum is then slowly released, and the pressure in the plating chamber is gradually returned to atmospheric conditions. The coated TMB is removed from the plating equipment. The surface morphology and thickness of the ZnO coating can be inspected using a microscope or scanning electron microscopy (SEM) to ensure uniformity and integrity. Additional tests such as adhesion testing, hardness testing, or electrical performance testing can be conducted to verify the functional properties of the ZnO coating.

### 2.2. Material and Experimental Method

The experiment used a selective laser melting machine, model S250. The internal structure of the working chamber of the equipment is shown in Figure 2a, with a printing area of 250 mm × 250 mm. The small substrate, measuring 100 mm × 100 mm, was installed in the center of the support plate. The distribution of the bolt base and signal lines below the substrate is shown in Figure 2b.

Table 1 lists the parameters of the S250 equipment. Table 2 lists the printing parameters used during the experiment.

The model used in the experiment was an H-shaped part, with the planar dimensions shown in Figure 3. The substrate in Figure 3a has dimensions of 250 mm × 250 mm and is referred to as the large substrate. The substrate in Figure 3b has dimensions of 100 mm × 100 mm and is referred to as the small substrate. Figure 2b shows the actual installation using the small substrate. The printed part is centered on the substrate, and its dimensions are a 40% proportional reduction in the part shown in Figure 3a. The thickness of both substrates used was 15 mm.

Two different substrate sizes were used in the experiments to verify the versatility of the monitoring system under different substrate dimensions and to compare the differences in signal values between these two conditions, exploring the relationship between signal magnitude and substrate size.

The scanning strategies used in the experiment are shown in Figure 4. Figure 4a illustrates the vertical bi-directional scanning strategy, where the scanning vector direction is vertical and the scanning sequence is from right to left. Figure 4b illustrates the horizontal bi-directional scanning strategy, where the scanning vector direction is horizontal and the scanning sequence is from bottom to top. Figure 4c illustrates the island scanning strategy, with red islands being scanned horizontally and blue islands being scanned vertically. Each island had dimensions of 15 mm × 15 mm, and the scanning order of the islands was a horizontal bi-directional scan from bottom to top. Figure 4d illustrates the rotational scanning strategy, where the scanning vector direction and sequence direction of each layer are rotated 90° clockwise from the previous layer. The first layer follows the scanning strategy shown in Figure 4a.

The experiments were divided into two groups. The first group of experiments used the small substrate configuration shown in Figure 3b and conducted four small experiments using the four scanning strategies depicted in Figure 4, studying the impact of different scanning strategies on deformation trend evolution. The second group of experiments used the structure shown in Figure 3a and conducted two small experiments using the scanning strategy shown in Figure 4a. The substrate was either not preheated or preheated to 200 °C to study the effect of preheating on deformation trend evolution.

### 2.3. Residual Stress and Deformation Development in AM

Understanding the evolution of residual stresses in the material and the reasons for part deformation during the LPBF process can help us comprehend the overall deformation trends of the part and the substrate during printing. Combining this with the signal value curves from the TMB Monitoring System can reveal the mechanisms of residual stress evolution in the part and explore the influence laws of different printing parameters and scan paths on the residual stresses and deformation of the part.

When a new layer of powder was spread on the powder bed and printed, the stress distribution near the laser-affected area is shown in Figure 5a. The laser-affected area expanded rapidly due to heating, but this expansion was constrained by the surrounding material, leading to significant compressive stress in this region, which exceeded its yield strength and caused plastic strain. The material in the lower layers was subjected to tensile forces from the upper layers, resulting in tensile stress and a tendency for the surrounding material to deform downward, creating a convex shape.

After the laser leaves the region, the area cooled rapidly in a short time, and the material contracted quickly. During the contraction process, it was constrained by the surrounding material. Since plastic strain had already occurred during the heating process, the material continued to contract after cooling to a stress equilibrium, evolving into tensile stress. The material in the lower layers was pulled inward, leading to compressive stress and a tendency for the surrounding material to deform upward, creating a concave shape.

Kruth et al. [10] provided a more detailed explanation, proposing the Temperature Gradient Mechanism (TGM). According to this mechanism, the rapid heating by the laser beam and the relatively slow heat conduction create a significant temperature gradient between the laser-affected area and the surrounding material, resulting in varying degrees of stress and strain within the material. Similar conclusions have been mentioned in the literature [9,15,29].

The laser acting on different areas at different times caused different deformations in the part and the substrate, following a specific pattern: the material in the laser-affected area deformed into a convex shape, causing downward displacement in the surrounding material, which resulted in a decrease in the signal values of nearby bolts. After the laser left, the material cooled and deformed into a concave shape, leading to a short-term increase in the signal values of nearby bolts. The closer the bolts were to the laser-affected area, the more pronounced this pattern became. Subsequent discussions of the bolt signal value curves are based on this pattern.

Before the printing began, the substrate was fixed with TMBs, and a certain pre-tension force was applied, with the bolt tension at this time reset to 0. After the printing started, the bolt tension began to change, and the output signal values represented the difference from the pre-tension force. Therefore, the signal values represented the magnitude of the deformation trend of the parts.

## 3. Results and Discussions

### 3.1. Signal Curves at Different Scanning Strategies

#### 3.1.1. Vertical Scanning Strategies on Small Substrate

Using a vertical scanning strategy with a small substrate for printing experiments, the bolt signal values collected were plotted as curves, as shown in Figure 6a. 

The entire experiment was divided into two stages: the printing stage and the cooling stage. During the printing stage, the signal values showed an increasing trend, while during the cooling stage, they showed a decreasing trend. The interval marked by the dashed line in Figure 6a includes the transition point between these two stages. The cycle for printing each layer is approximately 54 s, which includes about 14 s for powder spreading. A total of 99 layers were printed, and the printing ended at around 5360 s, after which the cooling stage began. Without considering the fluctuations, the four curves overall showed an increasing phase and a decreasing phase, with the increasing phase corresponding to the printing stage and the decreasing phase corresponding to the cooling stage, which aligned perfectly in time.

During the printing process, thermal stresses were generated in the part. After rapid cooling, these thermal stresses were not fully released, leading to residual stresses. In the layer-by-layer construction process, the cooling of each layer also affected the next layer, causing interlayer stress accumulation. This resulted in an overall increase in residual stresses and the cumulative deformation of the substrate. The deformation was restrained by the bolts, which converted it into tensile force in the bolts, reflected in the signal value magnitudes. The rate of increase in the curves gradually decreased with the increasing number of printed layers. This is because the first few layers are closer to the top surface of the substrate, and their residual stresses have a greater impact on substrate deformation; as the printing height increases, the influence of each layer’s residual stress on the substrate deformation decreases, leading to a slower increase in the bolt signal values.

The increasing phase of the curves shows periodic fluctuations. By counting the number of peaks, it is exactly 99, which corresponds to 99 cycles. Each cycle corresponds precisely to the time for printing each layer, indicating that the monitoring system has sufficient accuracy to present the stress changes in each layer during the printing process in a cyclic form.

During the cooling stage, the signal values decreased. This is because the part cools through natural convection, and as the temperature gradually decreases, some of the residual stresses are released, leading to a reduction in the bolt signal values.

During the printing process, the four bolts almost reached their maximum values at the same time. The sum of these maximum values was 13.21 kN, and the sum of the minimum values after cooling was 4.7 kN. The larger the two values, the greater the accumulative deformation trend.

The curve from 5200 s to 5500 s, which is the interval marked by the dashed line in Figure 6a, is shown in Figure 6b. All four curves exhibit periodic behavior. The signal value curves of bolts 1 and 2 are similar in shape, as are those of bolts 3 and 4. For vertical scanning, the temperature changes should be symmetrical up and down. Considering the bolt position distribution shown in Figure 3, bolts 1 and 2, and bolts 3 and 4 are symmetrically placed, so it is reasonable that their curve shapes are similar. The interval marked by the dashed line in Figure 6b represents the signal value changes within the time it takes to print one layer. By carefully observing the trends of increase and decrease, the following conclusions can be drawn:

When the laser acts on the right branch of the H-shaped part, according to the previous theoretical analysis, the area expands due to heat, forming a convex shape in the material (refer to Section 2.3). At this time, the tensile force of bolts 3 and 4 decreases rapidly. When the laser acts on the middle and left branches, the right branch cools down, transitioning from a convex shape to a concave shape, causing the tensile force of bolts 3 and 4 to gradually increase. Therefore, the signal values of bolts 3 and 4 first decrease rapidly and then increase gradually. Similarly, when the laser acts on the right branch, bolts 1 and 2 are less affected, and their signal values continue to rise slowly as the left branch cools. When the laser acts on the middle and left branches, the influence on bolts 1 and 2 gradually increases, causing their signal values to start decreasing. During the powder spreading stage, as the left branch cools, the signal values increase rapidly over a short time. Thus, the signal values of bolts 3 and 4 within one cycle first increase, then decrease, and then increase again.

It is observed that when the laser transitions from the right branch to the middle part, the signal value curves of bolts 3 and 4 reach a minimum, while the signal value curves of bolts 1 and 2 reach a maximum. When the laser acts on the middle part, the part as a whole assumes a convex shape, and all four bolts experience a reduction in tensile force. Therefore, the signal values of bolts 1 and 2 start to decrease slowly. During the powder spreading time, the left branch cools quickly, causing the signal values of bolts 1 and 2 to rise rapidly. However, when the laser moves from the right branch to the middle part, the signal values of bolts 3 and 4 do not rise as quickly, because the tensile force of all four bolts decreases when the laser acts on the middle part, which counteracts part of the increasing trend, resulting in a slower rise.

In Figure 6b, at 5360 s, which is after the third dashed line, the cooling stage begins. At the start of the cooling stage, the part transitions from a convex shape to a concave shape, causing a short-term increase in the signal values. After this initial phase, the part releases some of the residual stresses during the cooling process, leading to a long-term decrease in the signal values. Thus, the curves in the cooling stage first rise for a short time and then fall for a longer duration. Due to natural convection cooling, the rate of decrease is initially fast and then gradually slows down.

Figure 6c shows the physical part produced using the vertical scanning strategy on the small substrate. Careful observation reveals the vertical scanning tracks.

#### 3.1.2. Horizontal Scanning Strategies on Small Substrate

In the printing experiment using the vertical scanning strategy on a small substrate, the bolt signal values were collected and plotted as shown in Figure 7a.

The cycle time for printing each layer is approximately 102 s, including about 18 s for powder spreading. A total of 99 layers were printed, with the printing process ending at around 10,050 s, followed by the cooling stage.

The curve from 9800 s to 10,200 s is shown in Figure 7b. The signal value curves of bolts 1 and 4 are similar in shape, as are those of bolts 2 and 3. This is because the temperature changes are symmetrical left and right in the horizontal scanning strategy. By carefully observing the trends of increase and decrease, the following conclusions can be drawn: 

The scanning is from bottom to top, so when the laser acts on the lower part of the H-shaped part, the tensile force of bolts 2 and 3 decreases. When the laser acts on the middle and upper parts, the lower part cools, and the tensile force of bolts 2 and 3 gradually transitions from decreasing to increasing. Therefore, the signal values of bolts 2 and 3 first decrease and then increase. Similarly, when the laser acts on the lower part, bolts 1 and 4 are less affected, and their signal values continue to rise slowly. When the laser acts on the middle and upper parts, the influence on bolts 1 and 4 increases, causing their signal values to start decreasing. During the powder spreading stage, the signal values increase. Therefore, the signal values of bolts 1 and 4 within one cycle first increase, then decrease, and then increase again.

At the end of the printing, the sum of the signal values of the four bolts reached a maximum of 7.73 kN, while in the vertical scanning strategy, the maximum sum of the bolt signal values was 13.21 kN. This indicates that the deformation trend accumulation is smaller under the horizontal scanning strategy. This is related to the shape of the H-shaped part, as the average length of the scan vectors in the horizontal scanning strategy is shorter. As a result, adjacent scan lines are printed at temperatures significantly higher than the initial powder temperature, leading to a smaller thermal gradient and, consequently, less thermal stress accumulation [32].

Within one cycle, the signal value fluctuation (the difference between the maximum and minimum values within one cycle) under the horizontal scanning strategy is significantly smaller compared to the vertical scanning strategy. This is also because, in the vertical scanning strategy, the scan vectors for the left and right branches of the part are longer, resulting in greater thermal stress accumulation along the scan vectors [33], which leads to larger increases and decreases in the signal values.

During the cooling stage, the sum of the signal values eventually decreased to 2.03 kN, while in the vertical scanning strategy, the minimum value during the cooling stage was around 4.7 kN. This suggests that the accumulative deformation trend in the part under the horizontal scanning strategy is smaller than that under the vertical scanning strategy.

Figure 7c shows the physical part produced using the horizontal scanning strategy on the small substrate.

#### 3.1.3. Island Scanning Strategies on Small Substrate

In the printing experiment using the island scanning strategy on a small substrate, the bolt signal values were collected and plotted as shown in Figure 8a.

The cycle time for printing each layer is approximately 102 s, including about 18 s for powder spreading. A total of 99 layers were printed, with the printing process ending at around 10,030 s, followed by the cooling stage.

The curve from 9800 s to 10,200 s is shown in Figure 8b. All four curves exhibit a periodic behavior. The signal value curves of bolts 1 and 4 are similar in shape, as are those of bolts 2 and 3. Careful observation of the trends of increase and decrease yields conclusions similar to those under the horizontal scanning strategy: the scanning is from bottom to top. When the laser acts on the lower part of the H-shaped part, the tensile force of bolts 2 and 3 decreases. When the laser acts on the middle and upper parts, the lower part cools, and the tensile force of bolts 2 and 3 gradually increases. Therefore, the signal values of bolts 2 and 3 first decrease and then increase. Similarly, when the laser acts on the lower part, bolts 1 and 4 are less affected, and their signal values continue to rise slowly. When the laser acts on the middle and upper parts, the influence on bolts 1 and 4 increases, causing their signal values to start decreasing. During the powder spreading stage, the signal values increase. Therefore, the signal values of bolts 1 and 4 within one cycle first increase, then decrease, and then increase again.

The curve morphology within one cycle of island scanning is very similar to that of horizontal scanning. At the end of printing, the maximum sum of signal values under the island scanning strategy was 8.36 kN, and after cooling, the sum of signal values was 2.54 kN, both slightly larger than under the horizontal scanning strategy. This is because both scanning methods involve bidirectional horizontal scanning, but the island scanning strategy is conducted in units of islands. This scanning method causes the laser to act for a long time in a specific area, leading to excessively high temperatures in that region and sustained thermal stress accumulation, resulting in a larger deformation trend. When choosing scanning strategies in the future, it is important to avoid prolonging the laser’s action time in a specific area to prevent significant thermal stress accumulation. In subsequent experiments, islands will be scanned randomly rather than in sequence.

Figure 8c shows the physical part produced using the island scanning strategy on the small substrate. The contours of each island can be clearly seen.

#### 3.1.4. Rotational Scanning Strategies on Small Substrate

Rotational scanning: In the printing experiment using the rotational scanning strategy on a small substrate, the bolt signal values were collected and plotted as shown in Figure 9a.

Under this scanning strategy, the scan direction rotates 90° between adjacent layers, with every four layers forming one cycle. The time to print one layer in the horizontal direction is approximately 102 s, while the time to print one layer in the vertical direction is approximately 58 s, with a powder spreading time of 18 s. A total of 99 layers were printed, with the printing process ending at around 7990 s, followed by the cooling stage.

The curve from 7500 s to 8200 s is shown in Figure 9b. The blue dashed lines are used to divide the curve into four intervals, which correspond to the time periods of printing the last four layers. The scan directions for these four layers are as follows: Layer A is a horizontal scan from top to bottom, Layer B is a vertical scan from right to left, Layer C is a horizontal scan from bottom to top, and Layer D is a vertical scan from left to right. These four layers are labeled as Layers A, B, C, and D, respectively, and combined with the bolt numbers; the intervals with the same trends are marked with the same color.

Layer A is a horizontal scan from top to bottom. According to the previous analysis, under the horizontal scanning strategy, the trends of bolts 1 and 4 should be consistent, and similarly, the trends of bolts 2 and 3 should be consistent. This pattern is indeed observed in the figure. Layer C is a horizontal scan from bottom to top. Compared to Layer A, the only difference is the direction of the laser’s movement. Therefore, the trends of 1A and 4A should be consistent with those of 2C and 3C, marked in purple. Similarly, 1C and 4C are consistent with 2A and 3A, marked in blue.

At the end of printing, the maximum sum of the signal values of the four bolts was 5.92 kN, and the minimum value after cooling was 1.82 kN, which are the smallest among the four scanning strategies. This indicates that the rotational scanning method results in smaller deformation trend accumulation and residual stress accumulation [30,34]. Printing each layer in the same direction can lead to stress accumulation in a specific direction within the material. However, layer-by-layer rotational printing changes the print direction for each layer, thereby distributing the thermal stress in multiple directions and reducing the overall residual stress. This results in a more uniform stress distribution and reduces the risk of stress concentration.

Figure 9c shows the physical part produced using the rotational scanning strategy on the small substrate.

### 3.2. Signal Curves with and Without Preheating

#### 3.2.1. Vertical Scanning Strategies on Large Substrate

Vertical scanning on a large substrate: The signal value curve obtained from the experiment is shown in Figure 10a. 60 layers were printed, with each layer taking approximately 210 s, and the printing process ended at around 13,350 s.

It can be observed that in the large substrate experiment, the signal value trends are consistent with those of the small substrate. As shown in Figure 10b, the signal values of bolts 3 and 4 first rapidly decrease and then slowly increase within one cycle. For bolts 1 and 2, the signal values first increase, then decrease, and finally increase again within one cycle, with the first increase being very short or absent. This is because the large substrate has a larger area, and the time to print each layer is longer, with a larger cooling area. Therefore, when starting to print the next layer, the short-term increase in signal values is about to end.

Compared to the vertical scanning experiment on a small substrate, the fluctuations in the signal value curve within one cycle are not significantly different, but the magnitude of the signal values is significantly larger. The maximum signal value for the small substrate was in the range of 3–3.5 kN, while for the large substrate, it was in the range of 5–6 kN. This indicates that the larger the printing area, the greater the deformation of the substrate.

Figure 10c shows the physical part produced using the vertical scanning strategy on a large substrate. The part is larger in size, and the deformation is more noticeable, with significant warping observed on the left and right branches of the part. This is because the vertical scanning vectors for the branches are longer, leading to stress accumulation in the direction parallel to the vectors, causing noticeable warping in the vertical direction.

#### 3.2.2. Vertical Scanning Strategies on Large Preheating Substrate

Preheating experiment for large substrate vertical scanning: Before printing, the substrate was heated to 200 °C, and the other printing parameters were the same as in the previous experiment. Due to an occasional powder spreading defect during the printing process, a certain area experienced significant warping. A total of 30 layers were printed.

Figure 11a shows the signal value curve for the entire process. From 0 to 3000 s is the signal value change during the preheating phase, where the substrate expansion due to heating causes a certain tensile force on the bolts, leading to an upward trend in this segment of the curve. The time to print one layer is approximately 210 s, and the printing process ended at around 9500 s.

Figure 11b shows the extracted signal value curve from 8000 to 9000 s. Compared to the experiment without preheating, the trend of the signal values of bolts 1 and 2, which first increase, then decrease, and increase again within one cycle, is more pronounced in this experiment. This is because the substrate temperature is higher, and the cooling rate of the part is reduced, so the signal values are still in the rising phase when the next layer starts printing.

Compared to the experiment without preheating, the curve in the preheating experiment showed smaller fluctuations within a cycle, indicating a smaller temperature gradient in the printing area; the signal values at the same printing time were also smaller in the preheating experiment, indicating less deformation trend accumulation.

By preheating the substrate and the part, the temperature difference between the initial temperature and the laser heating area can be reduced. This results in a more gradual temperature gradient in the printing area, thereby reducing the generation of thermal stress. Preheating allows the material to start melting and cooling at a higher initial temperature, reducing the extent of local thermal deformation. This helps to reduce stress concentration caused by thermal deformation, thereby lowering residual stress.

In this experiment, it was observed that the lower end of the left branch of the H-shaped part experienced significant warping, as shown in Figure 11c. Observing the signal value curve in Figure 11a, it was found that the signal value of bolt 2 is significantly higher than that of the other three bolts. This leads to the conclusion that warping in the printed part reflects deformation in the substrate and can be detected by the TMB Monitoring System. Therefore, it is possible to design feedback control methods, such as adjusting printing parameters to correct the issue and continue printing after warping is detected, or directly stopping the machine to avoid material waste.

### 3.3. Limitations and Possible Errors in Experiments

The TMB Monitoring System has certain limitations. When the contact area between the printed part and the substrate is small, the changes in signal values are less significant. When there are numerous support structures between the part and the substrate, it can affect the accuracy of the monitoring results. During the multi-laser printing process, the changes in signal values become more complex, making them more difficult to analyze but still useful as a reference for comparing overall residual stress.

In these experiments, some sources of error can affect the accuracy and consistency of the results. For instance, non-uniform airflow distribution in the experimental equipment can lead to inconsistent print quality across different regions, causing unfused powder to fall back onto the part surface, and dust in the air leading to a reduction in the laser power illuminating the part, which can result in deviations in signal values. Additionally, variations in ambient temperature due to climate conditions and significant diurnal temperature differences may influence the signal values after cooling. The repeatability errors of the equipment, such as inconsistencies in powder spreading and oxygen content control, are also challenging to maintain uniformly, further contributing to signal value deviations. These factors can adversely affect the final print quality and residual stress analysis, thereby compromising the reliability and repeatability of the results.

## 4. Conclusions

A stress online monitoring system for metal additive manufacturing processes was designed in this study. The system uses TMBs as sensors, installed below the substrate, to reflect the deformation trend and its changes in the printed part in real time. Through two different experimental conditions, four different scanning path experiments, and two different preheating temperatures, the evolution mechanism of the deformation trend in the printed part was revealed, and the overall residual stress levels under different conditions were evaluated. The main conclusions are summarized as follows:The TMB stress online monitoring system proposed in this paper has high accuracy. The magnitude of the signal values can reflect the deformation trend of the part. It can accurately record the deformation changes in each layer during the printing process. The signal values after sufficient cooling can reflect the final overall residual stress level of the part.The material in the laser-affected area will expand thermally, causing the stretching of the surrounding material and downward displacement, resulting in a decrease in the tensile force of nearby bolts. When the laser leaves the area, the region cools down, causing the compression of the surrounding material and upward displacement; thus, the tensile force of nearby bolts increases in a short time.Different printing paths have different deformation trend accumulation processes. Among the four scanning strategies, the rotational scanning strategy has the smallest deformation trend. Interlayer rotation distributes thermal stresses in multiple directions, thereby reducing the overall residual stress. The vertical scanning strategy has the largest deformation trend, with longer scanning vectors causing thermal stresses to accumulate in the vector direction, leading to significant residual stress and warping. The deformation trend in island scanning is slightly higher than in horizontal scanning because island scanning concentrates more heat. When selecting scanning paths, it is recommended to avoid long scanning vectors and the prolonged heating of a single area.Using a preheated substrate during printing results in a small deformation trend in the printed parts compared to using an unheated substrate. After preheating, the temperature gradient between the printing area and the substrate decreases, helping to reduce the accumulation of thermal stress.

Through this study, we believe that in actual production processes, the TMB Monitoring System has the potential to accurately reflect the overall residual stress levels in parts by monitoring deformation trends. Currently, the system can be used to detect excessive thermal stresses that lead to warpage during the printing process, as well as to optimize process parameters for different parts and materials in order to reduce part overall deformation. We are conducting further experiments to establish a more precise relationship between signal values and overall residual stress levels.

## Figures and Tables

**Figure 1 sensors-25-01043-f001:**
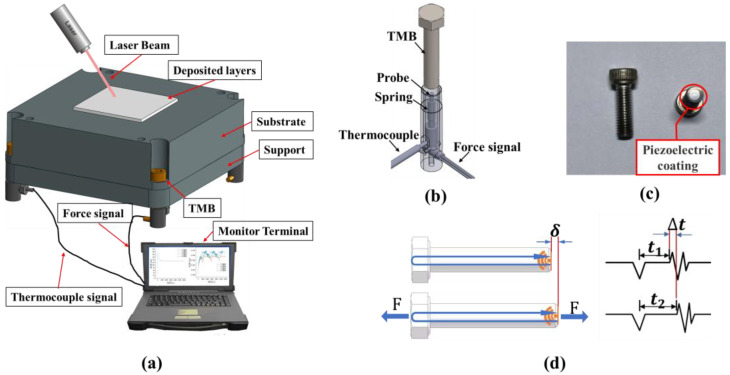
(**a**) Schematic diagram of the TMB Monitoring System. (**b**) Schematic diagram of the bolt base structure. (**c**) Photograph of the TMB. (**d**) Principal diagram of the TMB, The blue line on the left represents the path of the mechanical wave in the bolt, and the right side shows the corresponding waveform at the detection end..

**Figure 2 sensors-25-01043-f002:**
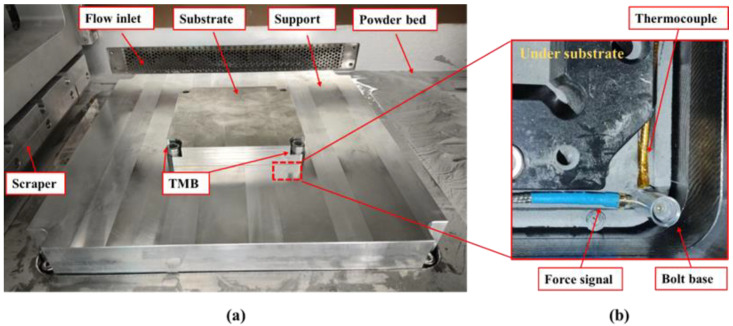
Photographs of the equipment and experimental environment: (**a**) the internal structure of the working chamber; (**b**) the bolt base and signal lines under the substrate.

**Figure 3 sensors-25-01043-f003:**
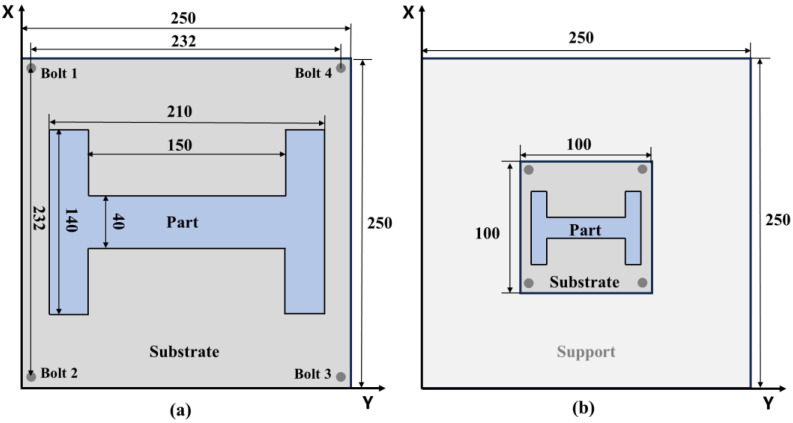
Schematic diagram of the printing working area, including the substrate, bolts, and printed parts: (**a**) schematic diagram of the structure using the large substrate, which covers the entire working area with dimensions of 250 mm × 250 mm; (**b**) schematic diagram of the structure using the small substrate, which is located in the center of the working area with dimensions of 100 mm × 100 mm. The printed part is scaled down proportionally.

**Figure 4 sensors-25-01043-f004:**
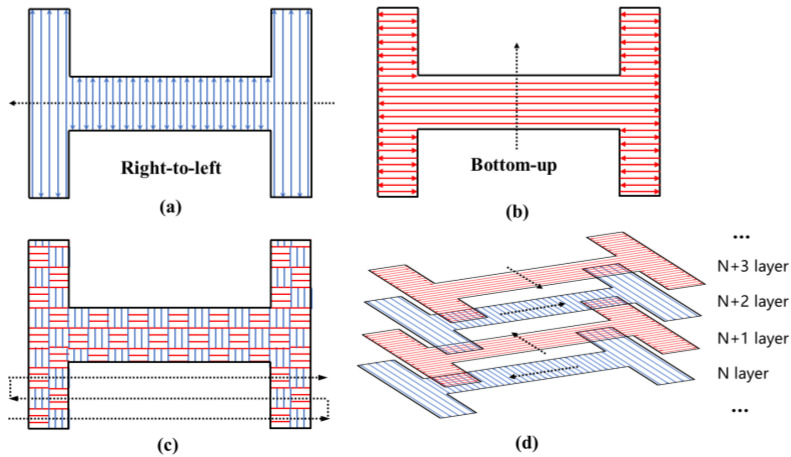
Schematic diagram of different scanning strategies used for the H-shaped part: (**a**) Vertical scanning strategy. The scanning vector direction is bi-directional vertical, and the scanning sequence is from right to left. (**b**) Horizontal scanning strategy. The scanning vector direction is bi-directional horizontal, and the scanning sequence is from bottom to top. (**c**) Island scanning strategy. The scanning vector directions of adjacent islands are perpendicular to each other. The scanning sequence is a bi-directional horizontal order from bottom to top. (**d**) Rotational scanning strategy. The scanning vector direction and sequence direction of each layer are rotated 90° clockwise from the previous layer. The first layer follows the scanning mode shown in Figure 4a.

**Figure 5 sensors-25-01043-f005:**
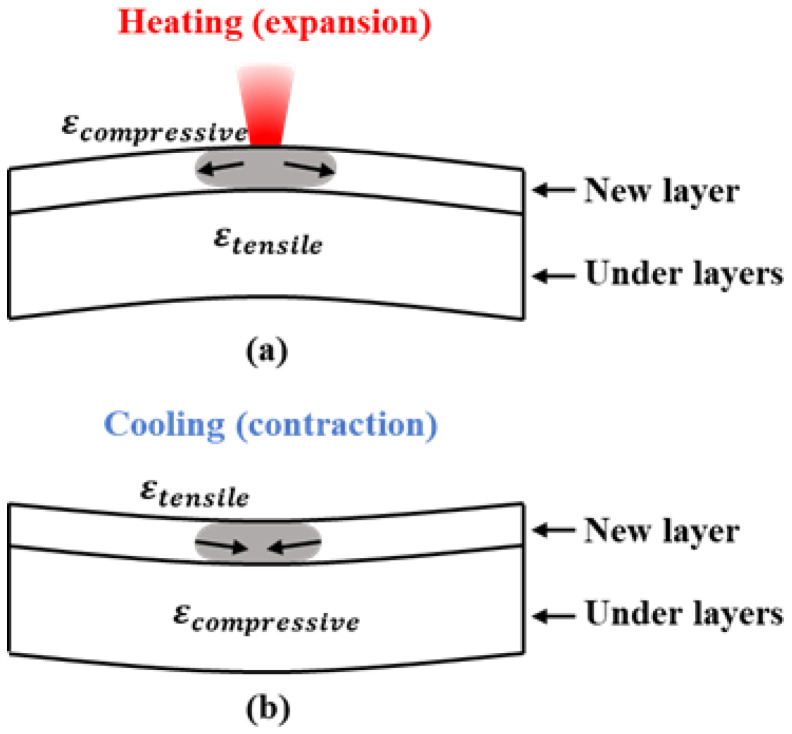
Mechanisms of material internal stress evolution near the laser-affected area during heating expansion and cooling contraction in the additive manufacturing process: (**a**) distribution of internal stresses in the material during the heating stage; (**b**) distribution of internal stresses in the material during the cooling stage. Based on [10].

**Figure 6 sensors-25-01043-f006:**
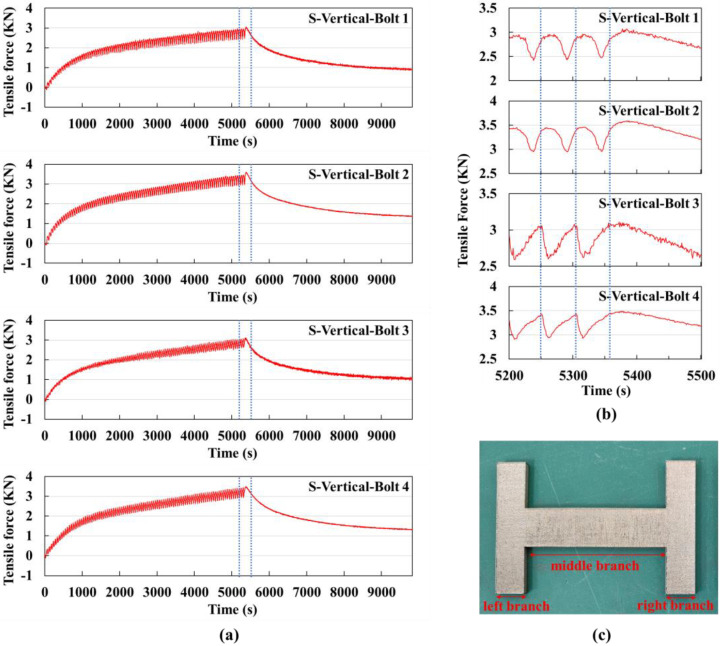
Vertical scanning strategy experiment using a small substrate: (**a**) Signal value curves of the four bolts for the entire experiment, with the dashed lines indicating the extracted time segment shown in (**b**) the signal value curve from 5200 s to 5500 s, where the dashed lines mark the time it takes to print one layer. (**c**) Physical image of the printed part.

**Figure 7 sensors-25-01043-f007:**
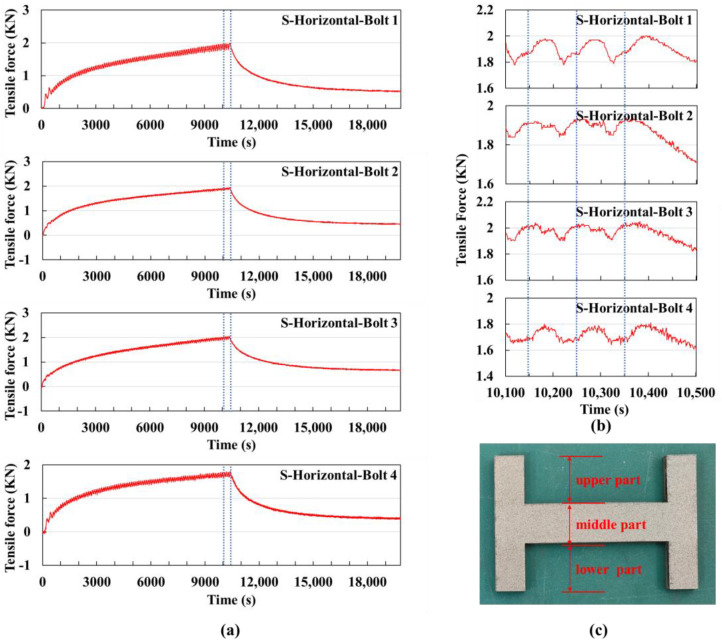
Vertical scanning strategy experiment using a small substrate: (**a**) Signal value curves of the four bolts for the entire experiment, with the dashed lines indicating the extracted time segment shown in (**b**) the signal value curve from 10,100 s to 10,500 s, where the dashed lines mark the time it takes to print one layer. (**c**) Physical image of the printed part.

**Figure 8 sensors-25-01043-f008:**
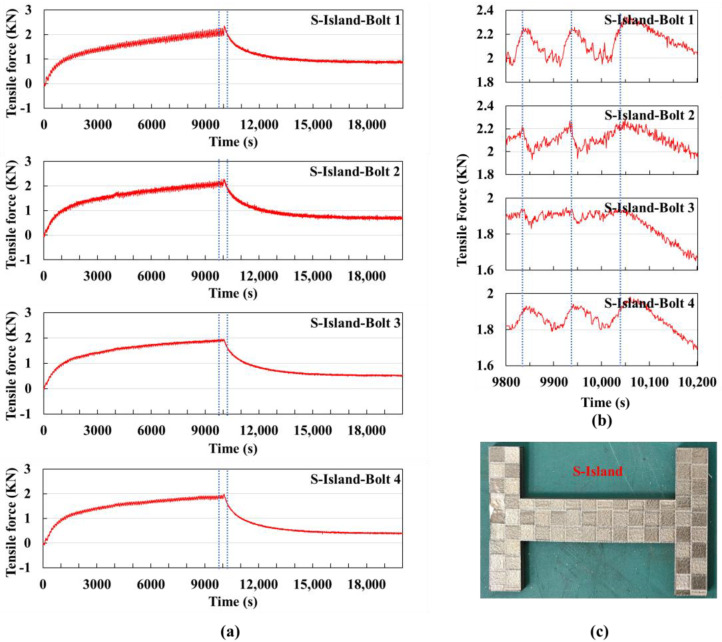
Island scanning strategy experiment using a small substrate: (**a**) Signal value curves of the four bolts for the entire experiment, with the dashed lines indicating the extracted time segment shown in (**b**) the signal value curve from 9800 s to 10,200 s, where the dashed lines mark the time it takes to print one layer. (**c**) Physical image of the printed part.

**Figure 9 sensors-25-01043-f009:**
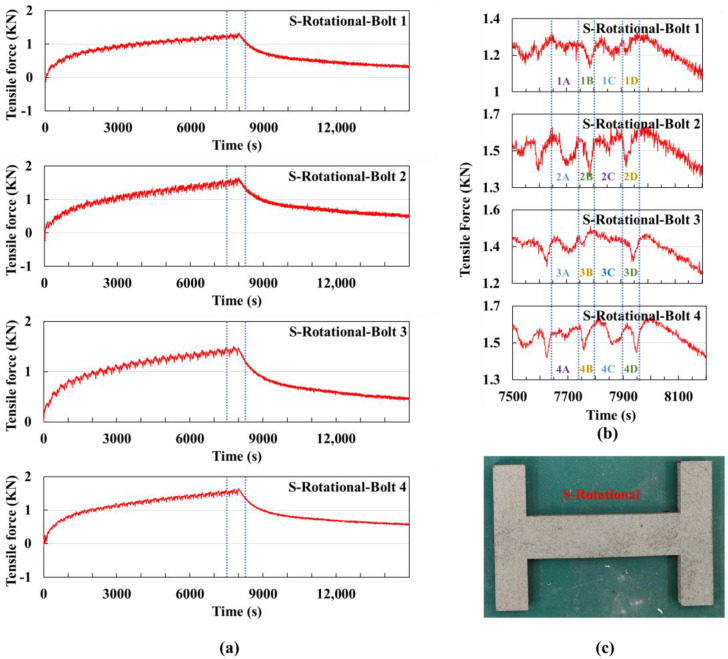
Rotational scanning strategy experiment using a small substrate: (**a**) Signal value curves of the four bolts for the entire experiment, with the dashed lines indicating the extracted time segment shown in (**b**) the signal value curve from 7500 s to 8200 s, where the dashed lines mark the time it takes to print one layer. (**c**) Physical image of the printed part.

**Figure 10 sensors-25-01043-f010:**
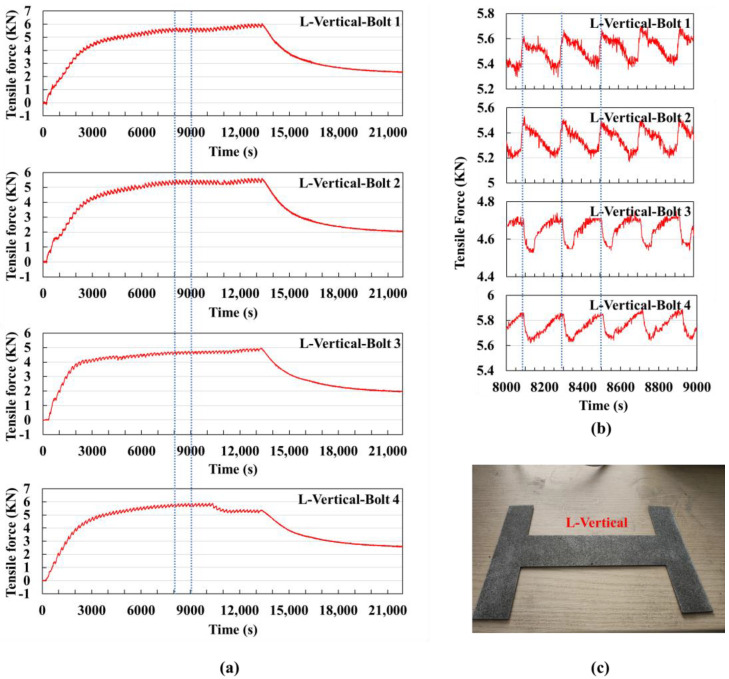
Vertical scanning strategy experiment using a large substrate without preheating: (**a**) Signal value curves of the four bolts for the entire experiment, with the dashed lines indicating the extracted time segment shown in (**b**) the signal value curve from 8000 s to 9000 s, where the dashed lines mark the time it takes to print one layer. (**c**) Physical image of the printed part.

**Figure 11 sensors-25-01043-f011:**
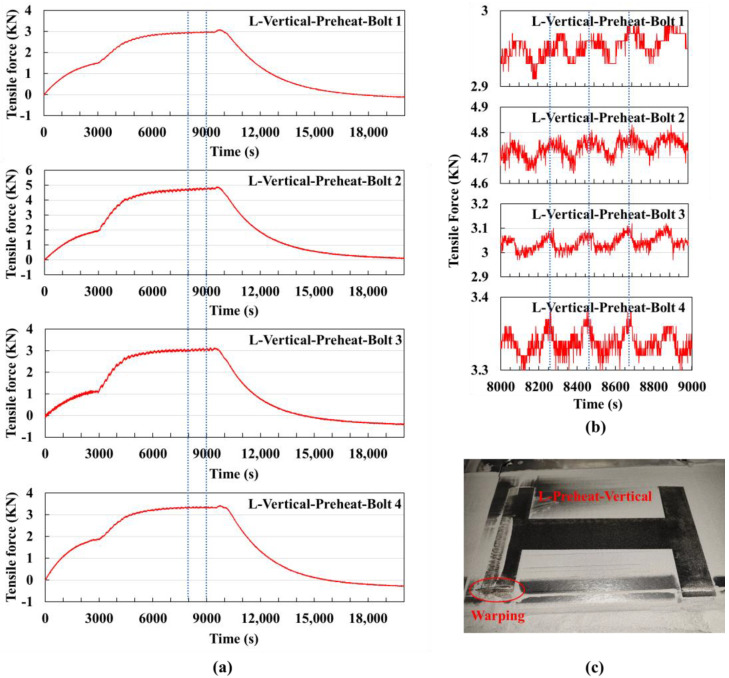
Vertical scanning strategy experiment using a large substrate preheated to 200 °C: (**a**) Signal value curves of the four bolts for the entire experiment, with the dashed lines indicating the extracted time segment shown in (**b**) the signal value curve from 8000 s to 9000 s, where the dashed lines mark the time it takes to print one layer. (**c**) Physical image of the printed part, showing significant warping deformation in the lower left corner.

**Table 1 sensors-25-01043-t001:** Main technical parameters of the S250.

Parameter	Value	Parameter	Value
Wavelength	1080 ± 5 nm	Hatch spacing	0.1 mm
Laser power	500 W	Layer thickness	0.02–0.05 mm
Scanning speed	≤10 m/s	Laser spot size	90 μm
Inert gas consumption	3–5 L/min	Fan working pressure	≤1000 Pa

**Table 2 sensors-25-01043-t002:** Processing parameters used in the present work.

Laser Power (W)	Laser Spot Size (μm)	Scan Speed (m/s)	Layer Thickness (μm)	Hatch Spacing (mm)
250	90	1	30	0.1

## Data Availability

The data presented in this study are available upon request from the corresponding author.

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
