# Peer review of "Study on the Evolution of Deformation Trend in Laser Powder Bed Fusion Samples Based on a Novel Monitoring System"

_sensors, 2025, doi:10.3390/s25041043_

Round 1

Reviewer 1 Report

Comments and Suggestions for Authors

This study presents an innovative technology to integrate a temperature sensor and force sensor by improving the bolt structure fixed to the printing substrate, to monitor the stress state of the substrate during the printing process. This technology not only shows innovation but also shows significant practical potential in application. It has an important reference value for scientific research and industrial practice in this domain. The topic of this study is highly compatible with the publication scope of the journal “Sensors”. 

Therefore, this study is recommended for publication. However, revisions and improvements are still needed before the formal publication.

In the derivation of Equations 1 – 4, the authors indicate that α can be neglected as a small quantity. However, in the actual derivation, α is retained in Equation 1 while α is neglected in Equation 4. Indeed, the degree of precision may not be sensitive to preserving α when calculating constant B. Overall, it is recommended that the authors maintain consistency in the derivation process, uniformly considering or ignoring the errors caused by α.

Why did the author choose the H-shaped block as the printing object? What is unique about this block in stress monitoring?

This study focuses on the residual stress, by monitoring the tension of the bolts around the substrate as an indicator of the substrate deformation, but is this monitoring too indirect to assess the residual stress of the deposited block itself? Was an analysis of the association between the detected tensile stress and the residual stress in the deposited block?

Can the author provide the monitored temperature of the bolts? Is the change in the electric signal caused by the temperature change fully considered? From Equation 1, if only considering the deformation difference caused by temperature change, is the error compensated enough, and does the piezoelectric constant also change synchronously?

Regarding the name of the sections, "3.1 Material and Experimental Method, 3.1.1 Vertical scanning strategies on small substrate" on page 10, the reviewer thinks that the authors may mistakenly use the wrong names.

The outlook of the conclusion section is ambitious, but it seems redundant.

Author Response

Comments 1: In the derivation of Equations 1 – 4, the authors indicate that α can be neglected as a small quantity. However, in the actual derivation, α is retained in Equation 1 while α is neglected in Equation 4. Indeed, the degree of precision may not be sensitive to preserving α when calculating constant B. Overall, it is recommended that the authors maintain consistency in the derivation process, uniformly considering or ignoring the errors caused by α.

Response 1: In equation 1, Δt and α are both small quantities and cannot be neglected. In equation 4, the factor 1+2αΔT, where a is much smaller compared to 1, can be neglected. The manuscript has already simplified equation 4. Line 180.

Comments 2: Why did the author choose the H-shaped block as the printing object? What is unique about this block in stress monitoring?

Response 2: The H-shaped part will provide richer variations in signal values compared to a regular rectangular shape. When the laser scans different areas, such as the left branch, middle, and right branch of the H-shape, it will produce different effects, which may help in discovering more details about the trend changes in the signal values. Apart from this, there is no other special significance.

Comments 3: This study focuses on the residual stress, by monitoring the tension of the bolts around the substrate as an indicator of the substrate deformation, but is this monitoring too indirect to assess the residual stress of the deposited block itself? Was an analysis of the association between the detected tensile stress and the residual stress in the deposited block?

Response3: From existing studies, it is observed that there is a relationship between the deformation of parts and residual stress. In this study, the signals obtained through direct measurement reflect the deformation trends of the substrate and the part. We also discuss the evolution of signal values from the perspectives of thermal stress and residual stress changes, proving that measuring deformation trends can, to some extent, reflect the overall stress levels in the part. However, direct evidence is lacking. For the sake of rigor, the manuscript has removed statements like “The TMB monitoring system can monitor the residual stress of parts” and instead emphasizes “The TMB monitoring system monitors the deformation trends of the part.” The precise relationship between bolt tension and part stress will be investigated in future research.

Comments 4: Can the author provide the monitored temperature of the bolts? Is the change in the electric signal caused by the temperature change fully considered? From Equation 1, if only considering the deformation difference caused by temperature change, is the error compensated enough, and does the piezoelectric constant also change synchronously?

Response 4: Sure, here’s the data from the longitudinal scan experiment with a small substrate in the attached document. The principle of a tension measuring bolt is to measure the transit time of mechanical waves traveling back and forth in the bolt shank, and then calculate the length change of the bolt. Therefore, the effect of temperature changes on the signal magnitude is primarily due to thermal expansion causing changes in the bolt’s length. Additionally, the impact of temperature changes on the speed of mechanical wave propagation can be largely ignored, and variations in the piezoelectric coefficient do not affect this measurement process because the strength of the electrical signal and mechanical waves does not influence the transit time signal. This is why the effect of temperature was not discussed in the manuscript. If there is a need to include a discussion of the temperature results, please let us know.

Comments 5: Regarding the name of the sections, "3.1 Material and Experimental Method, 3.1.1 Vertical scanning strategies on small substrate" on page 10, the reviewer thinks that the authors may mistakenly use the wrong names.

Response 5: Sorry, this is indeed a mistake. The section name has been changed. “3.1 Signal curves at different scanning strategies" on page 10, line 351.

Comments 6: The outlook of the conclusion section is ambitious, but it seems redundant.

Response 6:  The outlook has been deleted and replaced with a discussion on practical applications recommended by the other reviewer on page 21.

Reviewer 2 Report

Comments and Suggestions for Authors

The authors used Tension Measuring Bolt (TMB) to monitor the laser powder bed fusion substrate in real time and mapped the monitored data to stresses. This is an interesting piece of work, but the manuscript now requires modification and has the following issues that need to be answered by the authors.

1) In Section 3.1.1, the evolution in the vertical scanning experiment signal value curve are described using terms like ‘left branch’ and ‘right branch’. It is suggested to annotate these parts in Figure 6, which will facilitate the understanding of the readers.

2) Similarly, in Section 3.1.2, the evolution in the horizontal scanning experiment signal value curve are described using terms like upper part, middle part’, and lower part. It is suggested to annotate these parts in Figure 7.

3) It is recommended to discuss the limitations and possible errors in the experiments.

4) The signal value curves of symmetrically positioned bolts are similar in shape, but I suppose their values should be the same. For example, in Figure 6(b), the signal values of bolt 1 and bolt 2 differ by about 0.5 kN. Why is this the case? What impact might this have on the results?

5) Why use H-shaped parts? Have other shapes of parts been tried?

6) This article has excellent innovation and significance. It is recommended to emphasize the significance of the research work and how it can be applied in engineering production at the end of the article.

Author Response

Comments 1: In Section 3.1.1, the evolution in the vertical scanning experiment signal value curve are described using terms like ‘left branch’ and ‘right branch’. It is suggested to annotate these parts in Figure 6, which will facilitate the understanding of the readers.

Response 1: Thanks for your suggestion, related annotations have been added in Figure 6(c) on page 11.

Comments 2: Similarly, in Section 3.1.2, the evolution in the horizontal scanning experiment signal value curve are described using terms like upper partmiddle part’, and lower part. It is suggested to annotate these parts in Figure 7.

Response 2: Related annotations have been added in Figure 7(c) on page 13.

Comments 3: It is recommended to discuss the limitations and possible errors in the experiments.

Response 3: We have already added section "3.3 Limitations and possible errors in experiments", which describes the limitations and errors present in the experiments on page 20, line 628.

Comments 4: The signal value curves of symmetrically positioned bolts are similar in shape, but I suppose their values should be the same. For example, in Figure 6(b), the signal values of bolt 1 and bolt 2 differ by about 0.5 kN. Why is this the case? What impact might this have on the results?

Response 4: Ideally, the magnitude of the symmetric bolt signal values should be equal. However, during the experiment, there may be some errors, such as uneven wind fields causing unfused powder to fall back onto the part surface, and dust in the air leading to a reduction in the laser power illuminating the part. The powder spreading quality also has a certain degree of randomness, and inconsistencies in powder spreading quality across different regions can cause fluctuations in the signal values. These factors result in minor deviations in the symmetric bolt signal values, but the overall trend remains consistent.

Comments 5: Why use H-shaped parts? Have other shapes of parts been tried?

Response 5: The H-shaped part will provide richer variations in signal values compared to a regular rectangular shape. When the laser scans different areas, such as the left branch, middle, and right branch of the H-shape, it will produce different effects, which may help in discovering more details about the trend changes in the signal values. Of course, we also tried other shapes of parts, such as printing a rectangle or printing multiple small parts simultaneously. The results are consistent with the explanations in the manuscript.

Comments 6: This article has excellent innovation and significance. It is recommended to emphasize the significance of the research work and how it can be applied in engineering production at the end of the article.

Response 6: Thank you. We have add a discussion on significance and practical applications of the research at the end of the conclusion section on page 21, line 678.

Round 2

Reviewer 1 Report

Comments and Suggestions for Authors

The revised version of the paper responds well to the questions and comments previously made by the reviewer.

Regarding comment 4, it is appreciated that the authors provided a set of measurements of bolt temperatures at room temperature print. It was possible to see that the temperature varied very little over the measurement period. For the experiments that included substrate preheating, there should be some differences in the temperature ranges. However, the authors have done well in explaining how and why the measurements were taken, so this measurement method should be valid.

Hence, the paper is acceptable and recommended for publication.